

# Decentralized provenance-aware publishing with nanopublications

Tobias Kuhn[1], Christine Chichester[2], Michael Krauthammer[3,4], Núria Queralt-Rosinach[5], Ruben Verborgh[6], George Giannakopoulos[7,8], Axel-Cyrille Ngonga Ngomo[9], Raffaele Viglianti[10] and Michel Dumontier[11]

[1] Department of Computer Science, VU University Amsterdam, Amsterdam, Netherlands
[2] Nestle Institute of Health Sciences, Lausanne, Switzerland
[3] Yale University School of Medicine, Yale University, New Haven, CT, United States
[4] Yale Program in Computational Biology and Bioinformatics, Yale University, New Haven, CT, United States
[5] Research Programme on Biomedical Informatics, Hospital del Mar Medical Research Institute, Universitat Pompeu Fabra, Barcelona, Spain
[6] Data Science Lab, Ghent University, Ghent, Belgium
[7] Institute of Informatics and Telecommunications, NCSR Demokritos, Athens, Greece
[8] SciFY Private Not-for-profit Company, Athens, Greece
[9] AKSW Research Group, University of Leipzig, Leipzig, Germany
[10] Maryland Institute for Technology in the Humanities, University of Maryland, College Park, MD, United States
[11] Stanford Center for Biomedical Informatics Research, Stanford University, Stanford, CA, United States

Corresponding author
Tobias Kuhn, kuhntobias@gmail.com

## ABSTRACT

Publication and archival of scientific results is still commonly considered the responsability of classical publishing companies. Classical forms of publishing, however, which center around printed narrative articles, no longer seem well-suited in the digital age. In particular, there exist currently no efficient, reliable, and agreed-upon methods for publishing scientific datasets, which have become increasingly important for science. In this article, we propose to design scientific data publishing as a web-based bottom-up process, without top-down control of central authorities such as publishing companies. Based on a novel combination of existing concepts and technologies, we present a server network to decentrally store and archive data in the form of nanopublications, an RDF-based format to represent scientific data. We show how this approach allows researchers to publish, retrieve, verify, and recombine datasets of nanopublications in a reliable and trustworthy manner, and we argue that this architecture could be used as a low-level data publication layer to serve the Semantic Web in general. Our evaluation of the current network shows that this system is efficient and reliable.

# INTRODUCTION

Modern science increasingly depends on *datasets*, which are however left out in the classical way of publishing, i.e., through narrative (printed or online) articles in journals or conference proceedings. This means that the publications describing scientific findings become disconnected from the data they are based on, which can seriously impair the

verifiability and reproducibility of their results. Addressing this issue raises a number of practical problems: How should one publish scientific datasets and how can one refer to them in the respective scientific publications? How can we be sure that the data will remain available in the future and how can we be sure that data we find on the web have not been corrupted or tampered with? Moreover, how can we refer to specific entries or subsets from large datasets, for instance, to support a specific argument or hypothesis?

To address some of these problems, a number of scientific data repositories have appeared, such as Figshare and Dryad (http://figshare.com, http://datadryad.org). Furthermore, Digital Object Identifiers (DOI) have been advocated to be used not only for articles but also for scientific data (*Paskin, 2005*). While these approaches certainly improve the situation of scientific data, in particular when combined with Semantic Web techniques, they have nevertheless a number of drawbacks: they have *centralized* architectures, they give us no possibility to check whether the data have been (deliberately or accidentally) modified, and they do not support access or referencing on a more granular level than entire datasets (such as individual data entries). We argue that the centralized nature of existing data repositories is inconsistent with the decentralized manner in which science is typically performed, and that it has serious consequences with respect to reliability and trust. The organizations running these platforms might at some point go bankrupt, be acquired by investors who do not feel committed to the principles of science, or for other reasons become unable to keep their websites up and running. Even though the open licenses enforced by these data repositories will probably ensure that the datasets remain available at different places, there exist no standardized (i.e., automatable) procedures to find these alternative locations and to decide whether they are trustworthy or not.

Even if we put aside these worst-case scenarios, websites have typically not a perfect uptime and might be down for a few minutes or even hours every once in a while. This is certainly acceptable for most use cases involving a human user accessing data from these websites, but it can quickly become a problem in the case of automated access embedded in a larger service. Furthermore, it is possible that somebody gains access to the repository's database and silently modifies part of the data, or that the data get corrupted during the transfer from the server to the client. We can therefore never perfectly trust any data we get, which significantly complicates the work of scientists and impedes the potential of fully automatic analyses. Lastly, existing forms of data publishing have for the most part only one level at which data is addressed and accessed: the level of entire datasets (sometimes split into a small number of tables). It is in these cases not possible to refer to individual data entries or subsets in a way that is standardized and retains the relevant metadata and provenance information. To illustrate this problem, let us assume that we conduct an analysis using, say, 1,000 individual data entries from each of three very large datasets (containing, say, millions of data entries each). How can we now refer to exactly these 3,000 entries to justify whatever conclusion we draw from them? The best thing we can currently do is to republish these 3,000 data entries as a new dataset and to refer to the large datasets as their origin. Apart from the practical disadvantages of being forced to republish data just to refer to subsets of larger datasets, other scientists need to either (blindly) trust us or go through the tedious process of semi-automatically verifying that each of these entries

indeed appears in one of the large datasets. Instead of republishing the data, we could also try to describe the used subsets, e.g., in the form of SPARQL queries in the case of RDF data, but this doesn't make it less tedious, keeping in mind that older versions of datasets are typically not provided by public APIs such as SPARQL endpoints.

Below, we present an approach to tackle these problems, which builds upon existing Semantic Web technologies, in particular RDF and nanopublications, adheres to accepted web principles, such as decentralization and REST APIs, and supports the FAIR guiding principles of making scientific data Findable, Accessible, Interoperable, and Reusable (*Wilkinson et al., 2016*). Specifically, our research question is: Can we create a decentralized, reliable, and trustworthy system for publishing, retrieving, and archiving Linked Data in the form of sets of nanopublications based on existing web standards and infrastructure? It is important to note here that the word *trustworthy* has a broad meaning and there are different kinds of trust involved when it comes to retrieving and using datasets from some third party. When exploring existing datasets, a certain kind of trust is needed to decide whether an encountered dataset is appropriate for the given purpose. A different kind of trust is needed to decide whether an obtained file correctly represents a specific version of a specific dataset that has been chosen to be used. Only the second kind of trust can be achieved with a technical solution alone, and we use the word *trustworthy* in this paper in this narrow technical sense covering the second kind of trust.

This article is an extended and revised version of a previous conference paper (*Kuhn et al., 2015*). These extensions include, most importantly, a new evaluation on the retrieval of nanopublication datasets over an unreliable connection, a description of the new feature of surface patterns, the specific protocol applied by existing servers, a server network that is now three times as large as before (15 instead of five server instances), a much more detailed walk-through example, and five new figures. We furthermore present more details and discussions on topics including applications in the humanities, traversal-based querying, underspecified assertions, caching between architectural layers, and access of the server network via a web interface.

## BACKGROUND

Nanopublications (*Groth, Gibson & Velterop, 2010*) are a relatively recent proposal for improving the efficiency of finding, connecting, and curating scientific findings in a manner that takes attribution, quality levels, and provenance into account. While narrative articles would still have their place in the academic landscape, small formal data snippets in the form of nanopublications should take their central position in scholarly communication (*Mons et al., 2011*). Most importantly, nanopublications can be automatically interpreted and aggregated and they allow for fine-grained citation metrics on the level of individual claims. A nanopublication is defined as a small data container consisting of three parts: an assertion part containing the main content in the form of an atomic piece of formally represented data (e.g., an observed effect of a drug on a disease); a provenance part that describes how this piece of data came about (e.g., how it was measured); and a publication info part that gives meta-information about the nanopublication as a whole

(e.g., when it was created). The representation of a nanopublication with its three parts is based on the RDF language with named graphs (*Carroll et al., 2005*). In other words, the nanopublication approach boils down to the ideas of subdividing scientific results into atomic assertions, representing these assertions in RDF, attaching provenance information in RDF on the level of individual assertions, and treating each of these tiny entities as an individual publication. Nanopublications have been applied to a number of domains, so far mostly from the life sciences including pharmacology (*Williams et al., 2012*; *Banda et al., 2015*), genomics (*Patrinos et al., 2012*), and proteomics (*Chichester et al., 2015*). An increasing number of datasets formatted as nanopublications are openly available, including neXtProt (*Chichester et al., 2014*) and DisGeNET (*Queralt-Rosinach et al., 2015*), and the nanopublication concept has been combined with and integrated into existing frameworks for data discovery and integration, such as CKAN (*McCusker et al., 2013*).

Interestingly, the concept of nanopublications has also been taken up in the humanities, namely in philosophy (http://emto-nanopub.referata.com/wiki/EMTO_Nanopub), musicology (*Freedman, 2014*), and history/archaeology (*Golden & Shaw, 2016*). A humanities dataset of facts is arguably more interpretive than a scientific dataset; relying, as it does, on the scholarly interpretation of primary sources. Because of this condition, "facts" in humanities datasets (such as prosopographies) have often been called "factoids" (*Bradley, 2005*), as they have to account for a degree of uncertainty. Nanopublications, with their support for granular context and provenance descriptions, offer a novel paradigm for publishing such factoids, by providing methods for representing metadata about responsibilities and by enabling discussions and revisions beyond any single humanities project.

Research Objects are an approach related to nanopublications, aiming to establish "self-contained units of knowledge" (*Belhajjame et al., 2012*), and they constitute in a sense the antipode approach to nanopublications. We could call them *mega*-publications, as they contain much more than a typical narrative publication, namely resources like input and output data, workflow definitions, log files, and presentation slides. We demonstrate in this paper, however, that bundling all resources of scientific studies in large packages is not a necessity to ensure the availability of the involved resources and their robust interlinking, but we can achieve that also with cryptographic identifiers and a decentralized architecture.

SPARQL is an important and popular technology to access and publish Linked Data, and it is both a language to query RDF datasets (*Harris & Seaborne, 2013*) and a protocol to execute such queries on a remote server over HTTP (*Feigenbaum et al., 2013*). Servers that provide the SPARQL protocol, referred to as "SPARQL endpoints," are a technique for making Linked Data available on the web in a flexible manner. While off-the-shelf triple stores can nowadays handle billions of triples or more, they potentially require a significant amount of resources in the form of memory and processor time to execute queries, at least if the full expressive power of the SPARQL language is supported. A recent study found that more than half of the publicly accessible SPARQL endpoints are available less than 95% of the time (*Buil-Aranda et al., 2013*), posing a major problem to services depending on them, in particular to those that depend on several endpoints at the same time. To understand the consequences, imagine one has to program a mildly time-critical service that depends on RDF data from, say, ten different SPARQL endpoints. Assuming that each

endpoint is available 95% of the time and their availabilities are independent from each other, this means at least one of them will be down during close to five months per year. The reasons for this problem are quite clear: SPARQL endpoints provide a very powerful query interface that causes heavy load in terms of memory and computing power on the side of the server. Clients can request answers to very specific and complex queries they can freely define, all without paying a cent for the service. This contrasts with almost all other HTTP interfaces, in which the server imposes (in comparison to SPARQL) a highly limited interface, where the computational costs per request are minimal.

To solve these and other problems, more light-weight interfaces were suggested, such as the read-write Linked Data Platform interface (*Speicher, Arwe & Malhotra, 2015*), the Triple Pattern Fragments interface (*Verborgh et al., 2014*), as well as infrastructures to implement them, such as CumulusRDF (*Ladwig & Harth, 2011*). These interfaces deliberately allow less expressive requests, such that the maximal cost of each individual request can be bounded more strongly. More complex queries then need to be evaluated by clients, which decompose them in simpler subqueries that the interface supports (*Verborgh et al., 2014*). While this constitutes a scalability improvement (at the cost of, for instance, slower queries), it does not necessarily lead to perfect uptimes, as servers can be down for other reasons than excessive workload. We propose here to go one step further by relying on a *decentralized* network and by supporting only identifier-based lookup of nanopublications. Such limited interfaces normally have the drawback that traversal-based querying does not allow for the efficient and complete evaluation of certain types of queries (*Hartig, 2013*), but this is not a problem with the multi-layer architecture we propose below, because querying is only performed at a higher level where these limitations do not apply.

A well-known solution to the problem of individual servers being unreliable is the application of a decentralized architecture where the data is replicated on multiple servers. A number of such approaches related to data sharing have been proposed; for example, in the form of distributed file systems based on cryptographic methods for data that are public (*Fu, Kaashoek & Mazières, 2002*) or private (*Clarke et al., 2001*). In contrast to the design principles of the Semantic Web, these approaches implement their own internet protocols and follow the hierarchical organization of file systems. Other approaches build upon the existing BitTorrent protocol and apply it to data publishing (*Markman & Zavras, 2014*; *Cohen & Lo, 2014*), and there is interesting work on repurposing the proof-of-work tasks of Bitcoin for data preservation (*Miller et al., 2014*). There exist furthermore a number of approaches to applying peer-to-peer networks for RDF data (*Filali et al., 2011*), but they do not allow for the kind of permanent and provenance-aware publishing that we propose below. Moreover, only for the centralized and closed-world setting of database systems, approaches exist that allow for robust and granular references to subsets of dynamic datasets (*Proell & Rauber, 2014*).

The approach that we present below is based on previous work, in which we proposed *trusty URIs* to make nanopublications and their entire reference trees verifiable and immutable by the use of cryptographic hash values (*Kuhn & Dumontier, 2014*; *Kuhn & Dumontier, 2015*). This is an example of such a trusty URI:

```
http://example.org/r1.RA5AbXdpz5DcaYXCh9l3eI9ruBosiL5XDU3rxBbBaUO70
```

The last 45 characters of this URI (i.e., everything after " .") is what we call the *artifact code*. It contains a hash value that is calculated on the RDF content it represents, such as the RDF graphs of a nanopublication. Because this hash is part of the URI, any link to such an artifact comes with the possibility to verify its content, including other trusty URI links it might contain. In this way, the range of verifiability extends to the entire reference tree. Generating these trusty URIs does not come for free, in particular because the normalization of the content involves the sorting of the contained RDF statements. For small files such as nanopublications, however, the overhead is minimal, consisting only of about 1 millisecond per created nanopublication when the Java library is used (*Kuhn & Dumontier, 2014*; *Kuhn & Dumontier, 2015*).

Furthermore, we argued in previous work that the assertion of a nanopublication need not be fully formalized, but we can allow for informal or underspecified assertions (*Kuhn et al., 2013*), to deal with the fact that the creation of accurate semantic representations can be too challenging or too time-consuming for many scenarios and types of users. This is particularly the case for domains that lack ontologies and standardized terminologies with sufficient coverage. These structured but informal statements are supposed to provide a middle ground for the situations where fully formal statements are not feasible. We proposed a controlled natural language (*Kuhn, 2014*) for these informal statements, which we called AIDA (standing for the introduced restriction on English sentences to be atomic, independent, declarative, and absolute), and we had shown before that controlled natural language can also serve in the fully formalized case as a user-friendly syntax for representing scientific facts (*Kuhn et al., 2006*). We also sketched how "science bots" could autonomously produce and publish nanopublications, and how algorithms could thereby be tightly linked to their generated data (*Kuhn, 2015b*), which requires the existence of a reliable and trustworthy publishing system, such as the one we present here.

## APPROACH

Our approach on scientific data publishing builds upon the general Linked Data approach of lifting data on the web to linked RDF representations (*Berners-Lee, 2006*). We only deal here with structured data and assume that is already present in an RDF representation. The question of how to arrive at such a representation from other formats has been addressed by countless approaches—for example *Sequeda, Arenas & Miranker (2012)* and *Han et al. (2008)*—and is therefore outside of the scope of this paper. We furthermore exploit the fact that datasets in RDF can be split into small pieces without any effects on their semantics. After Skolemization of blank nodes, RDF triples are independent and can be separated and joined without restrictions. Best practices of how to define meaningful small groups of such triples still have to emerge, but an obvious and simple starting point is grouping them by the resource in subject position. We focus here on the technical questions and leave these practical issues for future research.

Specifically, our approach rests upon the existing concept of nanopublications and our previously introduced method of trusty URIs. It is a proposal of a reliable implementation of accepted Semantic Web principles, in particular of what has become known as the

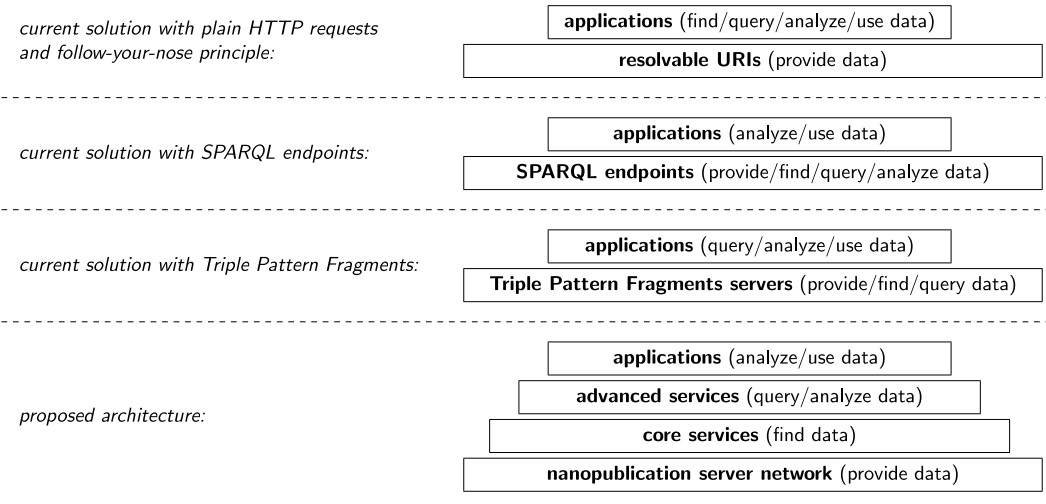

**Figure 1**  Illustration of current architectures of Semantic Web applications and our proposed approach.

*follow-your-nose* principle: Looking up a URI should return relevant data and links to other URIs, which allows one (i.e., humans as well as machines) to discover things by navigating through this data space (*Berners-Lee, 2006*). We argue that approaches following this principle can only be reliable and efficient if we have some sort of guarantee that the resolution of any single identifier will succeed within a short time frame in one way or another, and that the processing of the received representation will only take up a small amount of time and resources. This requires that (1) RDF representations are made available on several distributed servers, so the chance that they all happen to be inaccessible at the same time is negligible, and that (2) these representations are reasonably small, so that downloading them is a matter of fractions of a second, and so that one has to process only a reasonable amount of data to decide which links to follow. We address the first requirement by proposing a distributed server network and the second one by building upon the concept of nanopublications. Below we explain the general architecture, the functioning and the interaction of the nanopublication servers, and the concept of nanopublication indexes.

## Architecture

There are currently at least three possible architectures for Semantic Web applications (and mixtures thereof), as shown in a simplified manner in Fig. 1. The first option is the use of plain HTTP GET requests to dereference a URI. Applying the follow-your-nose principle, resolvable URIs provide the data based on which the application performs the tasks of finding relevant resources, running queries, analyzing and aggregating the results, and using them for the purpose of the application. This approach aligns very well with the principles and the architecture of the web, but the traversal-based querying it entails comes with limitations on efficiency and completeness (*Hartig, 2013*). If SPARQL endpoints are used, as a second option, most of the workload is shifted from the application to the server via the expressive power of the SPARQL query language. As explained above, this puts

servers at risk of being overloaded. With a third option such as Triple Pattern Fragments, servers provide only limited query features and clients perform the remainder of the query execution. This leads to reduced server costs, at the expense of longer query times.

We can observe that all these current solutions are based on two-layer architectures, and have moreover no inherent replication mechanisms. A single point of failure can cause applications to be unable to complete their tasks: A single URI that does not resolve or a single server that does not respond can break the entire process. We argue here that we need distributed and decentralized services to allow for robust and reliable applications that consume Linked Data. In principle, this can be achieved for any of these two-layer architectures by simply setting up several identical servers that mirror the same content, but there is no standardized and generally accepted way of how to communicate these mirror servers and how to decide on the client side whether a supposed mirror server is trustworthy. Even putting aside these difficulties, two-layer architectures have further conceptual limitations. The most low-level task of providing Linked Data is essential for all other tasks at higher levels, and therefore needs to be the most stable and robust one. We argue that this can be best achieved if we free this lowest layer from all tasks except the provision and archiving of data entries (nanopublications in our case) and decouple it from the tasks of providing services for finding, querying, or analyzing the data. This makes us advocate a multi-layer architecture, a possible realization of which is shown at the bottom of Fig. 1.

Below we present a concrete proposal of such a low-level data provision infrastructure in the form of a nanopublication server network. Based on such an infrastructure, one can then build different kinds of services operating on a subset of the nanopublications they find in the underlying network. "Core services" could involve things like resolving backwards references (i.e., "which nanopublications refer to the given one?") and the retrieval of the nanopublications published by a given person or containing a particular URI. Based on such core services for finding nanopublications, one could then provide "advanced services" that allow us to run queries on subsets of the data and ask for aggregated output. These higher layers can of course make use of existing techniques such as SPARQL endpoints and Triple Pattern Fragments or even classical relational databases, and they can cache large portions of the data from the layers below (as nanopublications are immutable, they are easy to cache). For example, an advanced service could allow users to query the latest versions of several drug-related datasets, by keeping a local triple store and providing users with a SPARQL interface. Such a service would regularly check for new data in the server network on the given topic, and replace outdated nanopublications in its triple store with new ones. A query request to this service, however, would *not* involve an immediate query to the underlying server network, in the same way that a query to the Google search engine does *not* trigger a new crawl of the web.

While the lowest layer would necessarily be accessible to everybody, some of the services on the higher level could be private or limited to a small (possibly paying) user group. We have in particular scientific data in mind, but we think that an architecture of this kind could also be used for Semantic Web content in general.

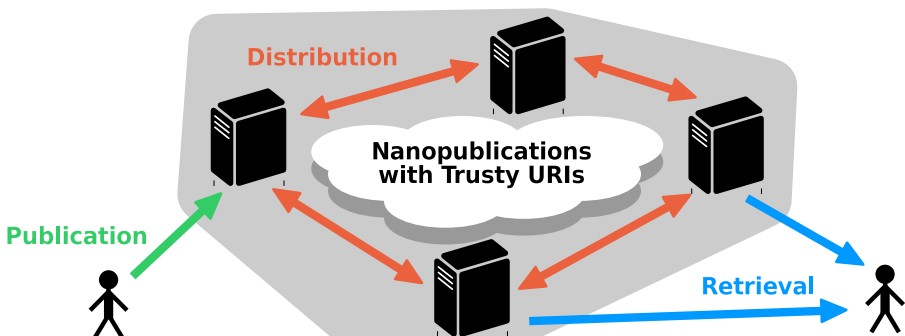

**Figure 2** **Schematic representation of the decentralized server architecture.** Nanopublications that have trusty URI identifiers can be uploaded to a server (or loaded from the local file system by the server administrator), and they are then distributed to the other servers of the network. They can then be retrieved from any of the servers, or from multiple servers simultaneously, even if the original server is not accessible.

## Nanopublication servers

As a concrete proposal of a low-level data provision layer, as explained above, we present here a decentralized nanopublication server network with a REST API to provide and distribute nanopublications. To ensure the immutability of these nanopublications and to guarantee the reliability of the system, these nanopublications are required to come with trusty URI identifiers, i.e., they have to be transformed on the client side into such trusty nanopublications before they can be published to the network. The nanopublication servers of such a network connect to each other to retrieve and (partly) replicate their nanopublications, and they allow users to upload new nanopublications, which are then automatically distributed through the network. Figure 2 shows a schematic depiction of this server network.

Basing the content of this network on nanopublications with trusty URIs has a number of positive consequences for its design: The first benefit is that the fact that nanopublications are always small (by definition) makes it easy to estimate how much time is needed to process an entity (such as validating its hash) and how much space to store it (e.g., as a serialized RDF string in a database). Moreover it ensures that these processing times remain mostly in the fraction-of-a-second range, guaranteeing that responses are always quick, and that these entities are never too large to be analyzed in memory. The second benefit is that servers do not have to deal with identifier management, as the nanopublications already come with trusty URIs, which are guaranteed to be unique and universal. The third and possibly most important benefit is that nanopublications with trusty URIs are immutable and verifiable. This means that servers only have to deal with *adding* new entries but not with *updating* them, which eliminates the hard problems of concurrency control and data integrity in distributed systems. (As with classical publications, a nanopublication—once published to the network—cannot be deleted or "unpublished," but only marked retracted or superseded by the publication of a new nanopublication.) Together, these aspects significantly simplify the design of such a network and its synchronization protocol, and make it reliable and efficient even with limited resources.

Specifically, a nanopublication server of the current network has the following components:

- A **key-value store** of its nanopublications (with the artifact code from the trusty URI as the key)
- A long list of all stored nanopublications, in the order they were loaded at the given server. We call this list the server's **journal**, and it consists of a journal identifier and the sequence of nanopublication identifiers, subdivided into pages of a fixed size. (1,000 elements is the default: page 1 containing the first 1,000 nanopublications; page 2 the next 1,000, etc.)
- A **cache of gzipped packages** containing all nanopublications for a given journal page
- **Pattern definitions** in the form of a *URI pattern* and a *hash pattern*, which define the surface features of the nanopublications stored on the given server
- A **list of known peers**, i.e., the URLs of other nanopublication servers
- **Information about each known peer**, including the journal identifier and the total number of nanopublications at the time it was last visited.

The server network can be seen as an unstructured peer-to-peer network, where each node can freely decide which other nodes to connect to and which nanopublications to replicate.

The URI pattern and the hash pattern of a server define the surface features of the nanopublications that this server cares about. We called them *surface features*, because they can be determined by only looking at the URI of a nanopublication. For example, the URI pattern '`http://rdf.disgenet.org/`' states that the given server is only interested in nanopublications whose URIs start with the given sequence of characters. Additionally, a server can declare a hash pattern like '`AA AB`' to state that it is only interested in nanopublications whose hash in the trusty URI start with one of the specified character sequences (separated by blank spaces). As hashes are represented in Base64 notation, this particular hash pattern would let a server replicate about 0.05% of all nanopublications. Nanopublication servers are thereby given the opportunity to declare which subset of nanopublications they replicate, and need to connect only to those other servers whose subsets overlap. To decide on whether a nanopublication belongs to a specified subset or not, the server only has to apply string matching at two given starting points of the nanopublication URI (i.e., the first position and position 43 from the end—as the hashes of the current version of trusty URIs are 43 bytes long), which is computationally cheap.

Based on the components introduced above, the servers respond to the following request (in the form of HTTP GET):

- Each server needs to return general **server information**, including the journal identifier and the number of stored nanopublications, the server's URI pattern and hash pattern, whether the server accepts POST requests for new nanopublications or servers (see below), and informative entries such as the name and email address of the maintainer and a general description. Additionally, some server-specific limits can be specified: the maximum number of triples per nanopublication (the default is 1,200), the

maximum size of a nanopublication (the default is 1 MB), and the maximum number of nanopublications to be stored on the given server (unlimited by default).

- Given an artifact code (i.e., the final part of a trusty URI) of a nanopublication that is stored by the server, it returns the given **nanopublication** in a format like TriG, TriX, N-Quads, or JSON-LD (depending on content negotiation).
- A **journal page** can be requested by page number as a list of trusty URIs.
- For every journal page (except for incomplete last pages), a gzipped **package** can be requested containing the respective nanopublications.
- The **list of known peers** can be requested as a list of URLs.

In addition, a server can optionally support the following two actions (in the form of HTTP POST requests):

- A server may accept requests to **add a given individual nanopublication** to its database.
- A server may also accept requests to **add the URL of a new nanopublication server** to its peer list.

Server administrators have the additional possibility to load nanopublications from the local file system, which can be used to publish large amounts of nanopublications, for which individual POST requests are not feasible.

Together, the server components and their possible interactions outlined above allow for efficient decentralized distribution of published nanopublications. Specifically, current nanopublication servers follow the following procedure.[1] Every server $s$ keeps its own list of known peer $P_s$. For each peer $p$ on that list that has previously been visited, the server additionally keeps the number of nanopublications on that peer server $n'_p$ and its journal identifier $j'_p$, as recorded during the last visit. At a regular interval, every peer server $p$ on the list of known peers is visited by server $s$:

1. The latest server information is retrieved from $p$, which includes its list of known peers $P_p$, the number of stored nanopublications $n_p$, the journal identifier $j_p$, the server's URI pattern $U_p$, and its hash pattern $H_p$.
2. All entries in $P_p$ that are not yet on the visiting server's own list of known peers $P_s$ are added to $P_s$.
3. If the visiting server's URL is not in $P_p$, the visiting server $s$ makes itself known to server $p$ with a POST request (if this is supported by $p$).
4. If the subset defined by the server's own URI/hash patterns $U_s$ and $H_s$ does not overlap with the subset defined by $U_p$ and $H_p$, then there won't be any nanopublications on the peer server that this server is interested in, and we jump to step 9.
5. The server will start at position $n$ to look for new nanopublications at server $p$: $n$ is set to the total number of nanopublications of the last visit $n'_p$, or to 0 if there was no last visit (nanopublication counting starts at 0).
6. If the retrieved journal identifier $j_p$ is different from $j'_p$ (meaning that the server has been reset since the last visit), $n$ is set to 0.
7. If $n = n_p$, meaning that there are no new nanopublications since the last visit, the server jumps to step 9.

[1] Source code repository: https://github.com/tkuhn/nanopub-server.

8. All journal pages $p$ starting from the one containing $n$ until the end of the journal are downloaded one by one (considering the size of journal pages, which is by default 1,000 nanopublications):

    (a) All nanopublication identifiers in $p$ (excluding those before $n$) are checked with respect to whether (A) they are covered by the visiting server's patterns $U_s$ and $H_s$ and (B) they are not already contained in the local store. A list $l$ is created of all nanopublication identifiers of the given page that satisfy both, (A) and (B).

    (b) If the number of new nanopublications $|l|$ exceeds a certain threshold (currently set to five), the nanopublications of $p$ are downloaded as a gzipped package. Otherwise, the new nanopublications (if any) are requested individually.

    (c) The retrieved nanopublications that are in list $l$ are validated using their trusty URIs, and all *valid* nanopublications are loaded to the server's nanopublication store and their identifiers are added to the end of the server's own journal. (Invalid nanopublications are ignored.)

9. The journal identifier $j_p$ and the total number of nanopublications $n_p$ for server $p$ are remembered for the next visit, replacing the values of $j'_p$ and $n'_p$.

The current implementation is designed to be run on normal web servers alongside with other applications, with economic use of the server's resources in terms of memory and processing time. In order to avoid overload of the server or the network connection, we restrict outgoing connections to other servers to one at a time. Of course, sufficient storage space is needed to save the nanopublications (for which we currently use MongoDB), but storage space is typically much easier and cheaper to scale up than memory or processing capacities. The current system and its protocol are not set in stone but, if successful, will have to evolve in the future—in particular with respect to network topology and partial replication—to accommodate a network of possibly thousands of servers and billions of nanopublications.

## Nanopublication indexes

To make the infrastructure described above practically useful, we have to introduce the concept of indexes. One of the core ideas behind nanopublications is that each of them is a tiny atomic piece of data. This implies that analyses will mostly involve more than just one nanopublication and typically a large number of them. Similarly, most processes will generate more than just one nanopublication, possibly thousands or even millions of them. Therefore, we need to be able to group nanopublications and to identify and use large collections of them.

Given the versatility of the nanopublication standard, it seems straightforward to represent such collections as nanopublications themselves. However, if we let such "collection nanopublications" contain other nanopublications, then the former would become very large for large collections and would quickly lose their property of being *nano*. We can solve part of that problem by applying a principle that we can call *reference instead of containment*: nanopublications cannot contain but only refer to other nanopublications, and trusty URIs allow us to make these reference links almost as strong as containment links. To emphasize this principle, we call them *indexes* and not collections.

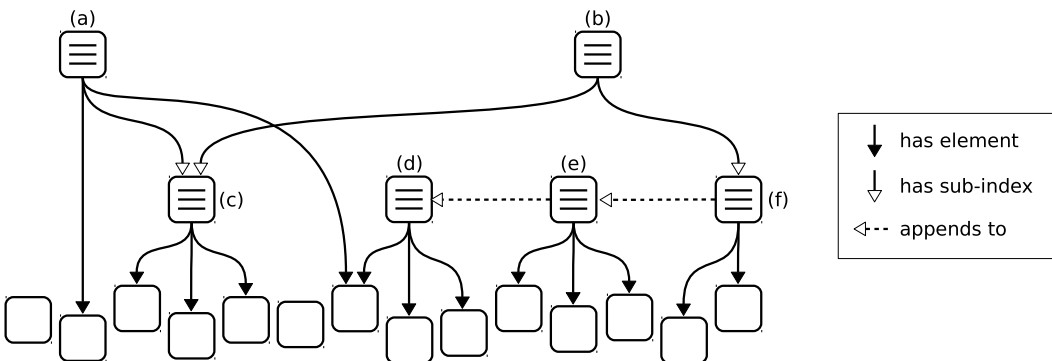

**Figure 3** **Schematic example of nanopublication indexes, which are themselves nanopublications.** Nanopublications can (but need not) be elements of one or more indexes. An index can have sub-indexes and can append to another index, in either case acquiring all nanopublications.

However, even by only containing references and not the complete nanopublications, these indexes can still become quite large. To ensure that all such index nanopublications remain *nano* in size, we need to put some limit on the number of references, and to support sets of arbitrary size, we can allow indexes to be appended by other indexes. We set 1,000 nanopublication references as the upper limit any single index can directly contain. This limit is admittedly arbitrary, but it seems to be a reasonable compromise between ensuring that nanopublications remain small on the one hand and limiting the number of nanopublications needed to define large indexes on the other. A set of 100,000 nanopublications, for example, can therefore be defined by a sequence of 100 indexes, where the first one stands for the first 1,000 nanopublications, the second one appends to the first and adds another 1,000 nanopublications (thereby representing 2,000 of them), and so on up to the last index, which appends to the second to last and thereby stands for the entire set. In addition, to allow datasets to be organized in hierarchies, we define that the references of an index can also point to sub-indexes. In this way we end up with three types of relations: an index can *append to* another index, it can contain other indexes as *sub-indexes*, and it can contain nanopublications as *elements*. These relations defining the structure of nanopublication indexes are shown schematically in Fig. 3. Index (a) in the shown example contains five nanopublications, three of them via sub-index (c). The latter is also part of index (b), which additionally contains eight nanopublications via sub-index (f). Two of these eight nanopublications belong directly to (f), whereas the remaining six come from appending to index (e). Index (e) in turn gets half of its nanopublications by appending to index (d). We see that some nanopublications may not be referenced by any index at all, while others may belong to several indexes at the same time. The maximum number of direct nanopublications (or sub-indexes) is here set to three for illustration purposes, whereas in reality this limit is set to 1,000.

In addition to describing sets of data entries, nanopublication indexes can also have additional metadata attached, such as labels, descriptions, further references, and other types of relations at the level of an entire dataset. Below we show how this general concept of

indexes can be used to define sets of new or existing nanopublications, and how such index nanopublications can be generated and published, and their nanopublications retrieved.

As a side note, dataset metadata can be captured and announced as nanopublications even for datasets that are not (yet) themselves available in the nanopublication format. The HCLS Community Profile of dataset descriptions (*Gray et al., 2015*) provides a good guideline of which of the existing RDF vocabularies to use for such metadata descriptions.

## Trusty publishing

Let us consider two simple exemplary scenarios to illustrate and motivate the general concepts. To demonstrate the procedure and the general interface of our implementation, we show here the individual steps on the command line in a tutorial-like fashion, using the np command from the `nanopub-java` library (*Kuhn, 2015a*). Of course, users should eventually be supported by graphical interfaces, but command line tools are a good starting point for developers to build such tools. To make this example completely reproducible, these are the commands to download and compile the needed code from a Bash shell (requiring Git and Maven):

```
$ git clone https://github.com/Nanopublication/nanopub-java.git
$ cd nanopub-java
$ mvn package
```

And for convenience reasons, we can add the *bin* directory to the path variable:

```
$ PATH=$(pwd)/bin:$PATH
```

To publish some new data, they have to be formatted as nanopublications. We use the TriG format here and define the following RDF prefixes:

```
@prefix : <http://example.org/np1#>.
@prefix xsd: <http://www.w3.org/2001/XMLSchema#>.
@prefix dc: <http://purl.org/dc/terms/>.
@prefix pav: <http://purl.org/pav/>.
@prefix prov: <http://www.w3.org/ns/prov#>.
@prefix np: <http://www.nanopub.org/nschema#>.
@prefix ex: <http://example.org/>.
```

A nanopublication consists of three graphs plus the head graph. The latter defines the structure of the nanopublication by linking to the other graphs:

```
:Head {
  : a np:Nanopublication;
    np:hasAssertion :assertion;
    np:hasProvenance :provenance;
    np:hasPublicationInfo :pubinfo.
}
```

The actual claim or hypothesis of the nanopublication goes into the assertion graph:

```
:assertion {
  ex:mosquito ex:transmits ex:malaria.
}
```

The provenance and publication info graph provide meta-information about the assertion and the entire nanopublication, respectively:

```
:provenance {
  :assertion prov:wasDerivedFrom ex:mypublication.
}
:pubinfo {
  : pav:createdBy <http://orcid.org/0000-0002-1267-0234>.
  : dc:created "2014-07-09T13:54:11+01:00"^^xsd:dateTime.
}
```

The lines above constitute a very simple but complete nanopublication. To make this example a bit more interesting, let us define two more nanopublications that have different assertions but are otherwise identical:

```
@prefix : <http://example.org/np2#>.
...
  ex:Gene1 ex:isRelatedTo ex:malaria.
...

@prefix : <http://example.org/np3#>.
...
  ex:Gene2 ex:isRelatedTo ex:malaria.
...
```

We save these nanopublications in a file `nanopubs.trig`, and before we can publish them, we have to assign them trusty URIs:

```
$ np mktrusty -v nanopubs.trig
Nanopub URI: http://example.org/np1#RAQoZlp22LHIvtYqHCosPbUtX8yeGs1Y5AfqcjMneLQ2I
Nanopub URI: http://example.org/np2#RAT5swlSLyMbuD03KzJsYHVV2oM1wRhluRxMrvpkZCDUQ
Nanopub URI: http://example.org/np3#RAkvUpysi9Ql3itlc6-iIJMG7YSt3-PI8dAJXcmafU71s
```

This gives us the file `trusty.nanopubs.trig`, which contains transformed versions of the three nanopublications that now have trusty URIs as identifiers, as shown by the output lines above. Looking into the file we can verify that nothing has changed with respect to the content, and now we are ready to publish them:

```
$ np publish trusty.nanopubs.trig
3 nanopubs published at http://np.inn.ac/
```

For each of these nanopublications, we can check their publication status with the following command (referring to the nanopublication by its URI or just its artifact code):

```
$ np status -a RAQoZlp22LHIvtYqHCosPbUtX8yeGs1Y5AfqcjMneLQ2I
URL: http://np.inn.ac/RAQoZlp22LHIvtYqHCosPbUtX8yeGs1Y5AfqcjMneLQ2I
Found on 1 nanopub server.
```

This is what you see immediately after publication. Only one server knows about the new nanopublication. Some minutes later, however, the same command leads to something like this:

```
$ np status -a RAQoZlp22LHIvtYqHCosPbUtX8yeGs1Y5AfqcjMneLQ2I
URL: http://np.inn.ac/RAQoZlp22LHIvtYqHCosPbUtX8yeGs1Y5AfqcjMneLQ2I
URL: http://ristretto.med.yale.edu:8080/nanopub-server/RAQoZlp22LHIvtYqHCosPbU...
URL: http://nanopubs.stanford.edu/nanopub-server/RAQoZlp22LHIvtYqHCosPbUtX8yeG...
URL: http://nanopubs.semanticscience.org:8082/RAQoZlp22LHIvtYqHCosPbUtX8yeGs1Y...
URL: http://rdf.disgenet.org/nanopub-server/RAQoZlp22LHIvtYqHCosPbUtX8yeGs1Y5A...
URL: http://app.tkuhn.eculture.labs.vu.nl/nanopub-server-2/RAQoZlp22LHIvtYqHCo...
URL: http://nanopubs.restdesc.org/RAQoZlp22LHIvtYqHCosPbUtX8yeGs1Y5AfqcjMneLQ2I
URL: http://nanopub.backend1.scify.org/nanopub-server/RAQoZlp22LHIvtYqHCosPbUt...
URL: http://nanopub.exynize.com/RAQoZlp22LHIvtYqHCosPbUtX8yeGs1Y5AfqcjMneLQ2I
Found on 9 nanopub servers.
```

Next, we can make an index pointing to these three nanopublications:

```
$ np mkindex -o index.nanopubs.trig trusty.nanopubs.trig
Index URI: http://np.inn.ac/RAXsXUhY8iDbfDdY6sm64hRFPr7eAwYXRlSsqQAz1LE14
```

This creates a local file `index.nanopubs.trig` containing the index, identified by the URI shown above. As this index is itself a nanopublication, we can publish it in the same way:

```
$ np publish index.nanopubs.trig
1 nanopub published at http://np.inn.ac/
```

Once published, we can check the status of this index and its contained nanopublications:

```
$ np status -r RAXsXUhY8iDbfDdY6sm64hRFPr7eAwYXRlSsqQAz1LE14
1 index nanopub; 3 content nanopubs
```

Again, after just a few minutes this nanopublication will be distributed in the network and available on multiple servers. From this point on, everybody can conveniently and reliably retrieve the given set of nanopublications. The only thing one needs to know is the artifact code of the trusty URI of the index:

```
$ np get -c RAXsXUhY8iDbfDdY6sm64hRFPr7eAwYXRlSsqQAz1LE14
```

This command downloads the nanopublications of the index we just created and published.

As another exemplary scenario, let us imagine a researcher in the biomedical domain who is interested in the protein CDKN2A and who has derived some conclusion based on the data found in existing nanopublications. Specifically, let us suppose this researcher analyzed the five nanopublications specified by the following artifact codes (they can be viewed online by appending the artifact code to the URL http://np.inn.ac/ or the URL of any other nanopublication server):

```
RAEoxLTy4pEJYbZwA9FuBJ6ogSquJobFitoFMbUmkBJh0
RAoMW0xMemwKEjCNWLFt8CgRmg_TGjfVSsh15hGfEmcz4
RA3BH_GncwEK_UXFGTvHcMVZ1hW775eupAccDdho5Tiow
RA3HvJ69nO0mD5d4m4u-Oc4bpXlxIWYN6L3wvB9jntTXk
RASx-fnzWJzluqRDe6GVMWFEyWLok8S6nTNkyElwapwno
```

These nanopublications about the same protein come from two different sources: The first one is from the BEL2nanopub dataset, whereas the remaining four are from neXtProt. (See https://github.com/tkuhn/bel2nanopub and http://nextprot2rdf.sourceforge.net, respectively, and Table 1.) These nanopublications can be downloaded as above with the `np get` command and stored in a file, which we name here `cdkn2a-nanopubs.trig`.

In order to be able to refer to such a collection of nanopublications with a single identifier, a new index is needed that contains just these five nanopublications. This time we give the index a title (which is optional).

```
$ np mkindex -t "Data about CDKN2A from BEL2nanopub & neXtProt" \
  -o index.cdkn2a-nanopubs.trig cdkn2a-nanopubs.trig
Index URI: http://np.inn.ac/RA6jrrPL2NxxFWlo6HFWas1ufp0OdZzS_XKwQDXpJg3CY
```

The generated index is stored in the file `index.cdkn2a-nanopubs.trig`, and our exemplary researcher can now publish this index to let others know about it:

```
$ np publish index.cdkn2a-nanopubs.trig
1 nanopub published at http://np.inn.ac/
```

**Table 1  Existing datasets in the nanopublication format, five of which were used for the first part of the evaluation.**

| Dataset | # nanopublications | | # triples | | Used for evaluation part |
|---|---|---|---|---|---|
| *Name and citation* | *Index* | *Content* | *Index* | *Content* | |
| GeneRIF/AIDA | 157 | 156,026 | 157,909 | 2,340,390 | First |
| (*NP Index* RAY_lQruua, *2015*) | | | | | |
| OpenBEL 1.0 | 53 | 50,707 | 51,448 | 1,502,574 | First |
| (*NP Index* RACy0I4f_w, *2015*) | | | | | |
| OpenBEL 20131211 | 76 | 74,173 | 75,236 | 2,186,874 | First |
| (*NP Index* RAR5dwELYL, *2015*) | | | | | |
| DisGeNET v2.1.0.0 | 941 | 940,034 | 951,325 | 31,961,156 | First |
| (*NP Index* RAXy332hxq, *2015*) | | | | | |
| DisGeNET v3.0.0.0 | 1,019 | 1,018,735 | 1,030,962 | 34,636,990 | None |
| (*NP Index* RAVEKRW0m6, *2015*) | | | | | |
| neXtProt | 4,026 | 4,025,981 | 4,078,318 | 156,263,513 | First |
| (*NP Index* RAXFlG04YM, *2015*) | | | | | |
| LIDDI | 99 | 98,085 | 99,272 | 2,051,959 | Third |
| (*NP Index* RA7SuQ0e66, *2015*) | | | | | |
| *Total* | 6,371 | 6,363,741 | 6,444,470 | 230,943,456 | |

There is no need to publish the five nanopublications this index is referring to, because they are already public (this is how we got them in the first place). The index URI can now be used to refer to this new collection of existing nanopublications in an unambiguous and reliable manner. This URI can be included in the scientific publication that explains the new finding, for example with a reference like the following:

[1] Data about CDKN2A from BEL2nanopub & neXtProt. Nanopublication index http://np.inn.ac/RA6jrrPL2NxxFWlo6HFWas1ufp0OdZzS_XKwQDXpJg3CY, 14 April 2015.

In this case with just five nanopublications, one might as well refer to them individually, but this is obviously not an option for cases where we have hundreds or thousands of them. The given web link allows everybody to retrieve the respective nanopublications via the server np.inn.ac. The URL will not resolve should the server be temporarily or permanently down, but because it is a trusty URI we can retrieve the nanopublications from any other server of the network following a well-defined protocol (basically just extracting the artifact code, i.e., the last 45 characters, and appending it to the URL of another nanopublication server). This reference is therefore much more reliable and more robust than links to other types of data repositories. In fact, we refer to the datasets we use in this publication for evaluation purposes, as described below in 'Evaluation', in exactly this way (*NP Index* RAY_lQruua, *2015*; *NP Index* RACy0I4f_w, *2015*; *NP Index* RAR5dwELYL, *2015*; *NP Index* RAXy332hxq, *2015*; *NP Index* RAVEKRW0m6, *2015*; *NP Index* RAXFlG04YM, *2015*; *NP Index* RA7SuQ0e66, *2015*).

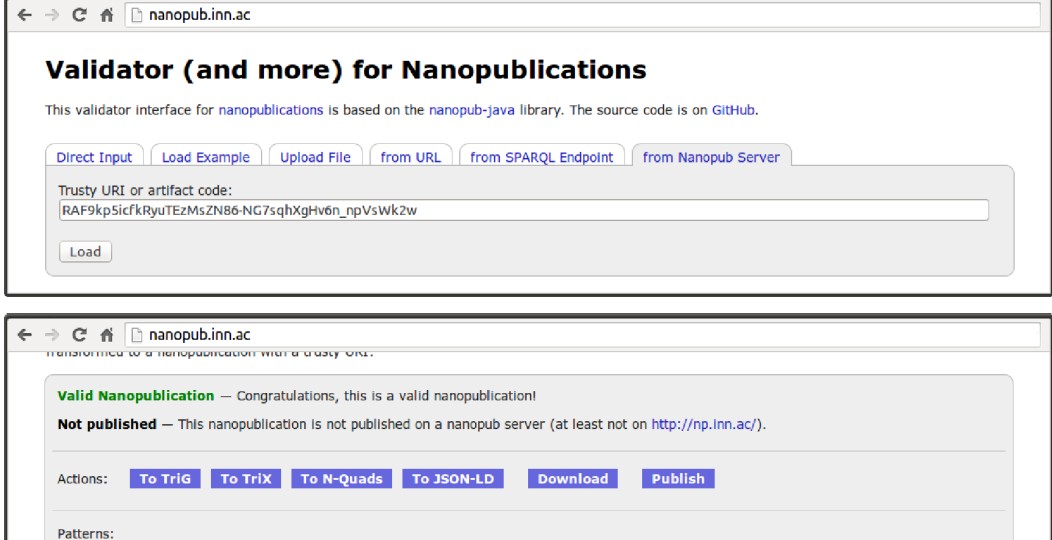

**Figure 4** **The web interface of the nanopublication validator can load nanopublications by their trusty URI (or just their artifact code) from the nanopublication server network.** It also allows users to directly publish uploaded nanopublications.

The new finding that was deduced from the given five nanopublications can, of course, also be published as a nanopublication, with a reference to the given index URI in the provenance part:

```
@prefix : <http://example.org/myfinding#>.
...
@prefix nps: <http://np.inn.ac/>.
@prefix uniprot: <http://purl.uniprot.org/uniprot/>.
...
:assertion {
  uniprot:P42771 a ex:proteinWithPropertyX.
}
:provenance {
  :assertion prov:wasInfluencedBy
      nps:RA6jrrPL2NxxFWlo6HFWas1ufp0OdZzS_XKwQDXpJg3CY.
}
:pubinfo {
  : pav:createdBy <http://orcid.org/0000-0002-1267-0234>.
  : dc:created "2015-04-14T08:05:43+01:00"^^xsd:dateTime.
}
```

We can again transform it to a trusty nanopublication , and then publish it as above.

Some of the features of the presented command-line interface are made available through a web interface for dealing with nanopublications that is shown in Fig. 4. The supported features include the generation of trusty URIs, as well as the publication and retrieval of nanopublications. The interface allows us to retrieve, for example, the nanopublication we just generated and published above, even though we used an `example.org` URI, which is not directly resolvable. Unless it is just about toy examples, we should of course try to use

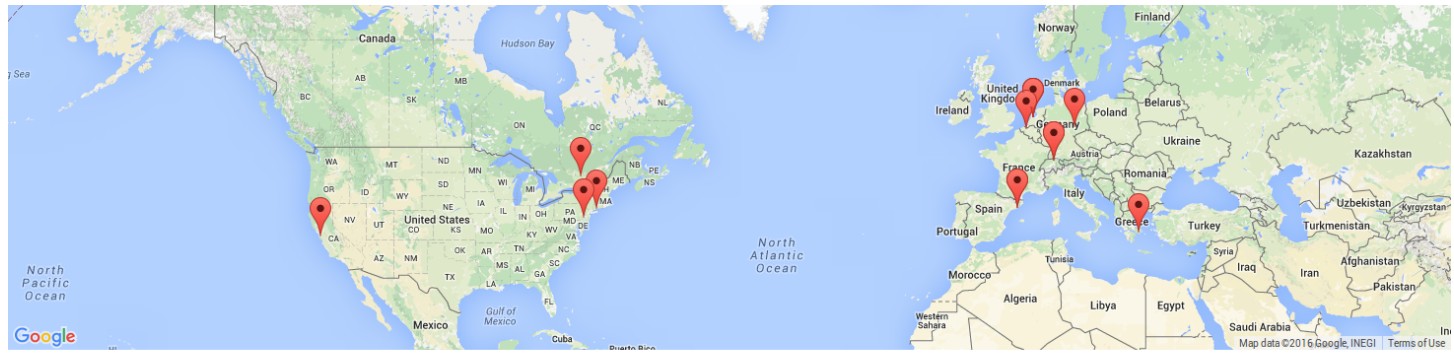

| URL | Status | OK Ratio | Resp Time (Dist) | Last Seen OK | NP Count | IP Address | Server Location | Version | URI / Hash Pattern |
|---|---|---|---|---|---|---|---|---|---|
| http://ristretto.med.yale.edu:8080/nanopub-server/ | OK | 99.98538% | 195 ms (6212 km) | 2016-02-05 08:06:23 | 6370147 | 128.36.40.86 | New Haven, United States | 0.2 | / |
| http://np.inn.ac/ | OK | 99.99707% | 29 ms (0 km) | 2016-02-05 08:06:27 | 6370147 | 129.132.255.27 | Zurich, Switzerland | 0.3 | / |
| http://npx1.inn.ac/ | OK | 100.0% | 6 ms (0 km) | 2016-02-05 08:06:26 | 398634 | 129.132.255.27 | Zurich, Switzerland | 0.3 | / A B C D |
| http://app.tkuhn.eculture.labs.vu.nl/nanopub-server-1/ | OK | 99.970764% | 33 ms (615 km) | 2016-02-05 08:06:26 | 1593807 | 130.37.193.11 | Amsterdam, Netherlands | 0.3 | / A B C D E F G H I J K L M N O P |
| http://app.tkuhn.eculture.labs.vu.nl/nanopub-server-2/ | OK | 99.98538% | 54 ms (615 km) | 2016-02-05 08:06:24 | 1592080 | 130.37.193.11 | Amsterdam, Netherlands | 0.3 | / Q R S T U V W X Y Z a b c d e f |
| http://app.tkuhn.eculture.labs.vu.nl/nanopub-server-3/ | OK | 99.988304% | 32 ms (615 km) | 2016-02-05 08:06:24 | 1592191 | 130.37.193.11 | Amsterdam, Netherlands | 0.3 | / g h i j k l m n o p q r s t u v |
| http://app.tkuhn.eculture.labs.vu.nl/nanopub-server-4/ | OK | 99.97953% | 34 ms (615 km) | 2016-02-05 08:06:27 | 1592069 | 130.37.193.11 | Amsterdam, Netherlands | 0.3 | / w x y z 0 1 2 3 4 5 6 7 8 9 - _ |
| http://nanopubs.semanticscience.org:8082/ | OK | 99.94445% | 215 ms (6131 km) | 2016-02-05 08:06:22 | 6370147 | 134.117.221.11 | Ottawa, Canada | 0.4 | / |
| http://nanopub.exynize.com/ | OK | 100.0% | 49 ms (516 km) | 2016-02-05 08:06:23 | 6370147 | 139.18.2.164 | Leipzig, Germany | 0.4 | / |
| http://nanopub.backend1.scify.org/nanopub-server/ | OK | 99.9775% | 185 ms (1616 km) | 2016-02-05 08:06:27 | 6370147 | 143.233.226.41 | Athens, Greece | 0.4 | / |
| http://nanopubs.restdesc.org/ | OK | 99.99707% | 66 ms (539 km) | 2016-02-05 08:06:22 | 6370147 | 157.193.213.74 | Ghent, Belgium | 0.3 | / |
| http://digitalduchemin.org/np-mirror/ | OK | 99.0991% | 196 ms (6460 km) | 2016-02-05 08:06:21 | 199421 | 165.82.124.16 | Haverford, United States | 0.4 | / E F |
| http://nanopubs.stanford.edu/nanopub-server/ | OK | 100.0% | 375 ms (9394 km) | 2016-02-05 08:06:25 | 6370147 | 171.67.213.57 | Stanford, United States | 0.2 | / |
| http://rdf.disgenet.org/nanopub-server-disgenet/ | OK | 99.994156% | 57 ms (835 km) | 2016-02-05 08:06:26 | 1959788 | 84.89.134.134 | Barcelona, Spain | 0.3 | http://rdf.disgenet.org/ / |
| http://rdf.disgenet.org/nanopub-server/ | OK | 99.988304% | 90 ms (835 km) | 2016-02-05 08:06:24 | 6370147 | 84.89.134.134 | Barcelona, Spain | 0.2 | / |

**Figure 5**  **This screenshot of the nanopublication monitor interface (http://npmonitor.inn.ac ) showing the current server network.** It currently consists of 15 server instances on 10 physical servers in Zurich, New Haven, Ottawa, Amsterdam, Stanford, Barcelona, Ghent, Athens, Leipzig, and Haverford.

resolvable URIs, but with our decentralized network we can retrieve the data even if the original link is no longer functioning or temporarily broken.

## EVALUATION

To evaluate our approach, we want to find out whether a small server network run on normal web servers, without dedicated infrastructure, is able to handle the amount of nanopublications we can expect to become publicly available in the next few years. Our evaluation consists of three parts focusing on the different aspects of dataset publication, server performance, and dataset retrieval, respectively. At the time the first part of the evaluation was performed, the server network consisted of three servers in Zurich, New Haven, and Ottawa. Seven new sites in Amsterdam, Stanford, Barcelona, Ghent, Athens, Leipzig, and Haverford have joined the network since. The current network of 15 server instances on 10 sites (in 8 countries) is shown in Fig. 5, which is a screenshot of a nanopublication monitor that we have implemented (https://github.com/tkuhn/nanopub-monitor). Such monitors regularly check the nanopublication server network, register changes (currently once per minute), and test the response times and the correct operation of the servers by requesting a random nanopublication and verifying the returned data.

The files of the presented studies are available online in two repositories, one for the analyses of the original studies that have been previously published (https://bitbucket.org/tkuhn/trustypublishing-study/) and another one with the files for the additional analyses and diagrams for this extended article (https://bitbucket.org/tkuhn/trustypublishingx-study/).

## Evaluation design

Table 1 shows seven existing large nanopublication datasets. Five of these datasets were used for the first part of the evaluation (the other two were not yet available at the time this part of the evaluation was conducted), which tested the ability of the network to store and distribute new datasets. These five datasets consist of a total of more than 5 million nanopublications and close to 200 million RDF triples, including nanopublication indexes that we generated for each dataset. The total size of these five datasets when stored as uncompressed TriG files amounts to 15.6 GB. Each of the datasets is assigned to one of the three servers, where it is loaded from the local file systems. The first nanopublications start spreading to the other servers, while others are still being loaded from the file system. We therefore test the reliability and capacity of the network under constant streams of new nanopublications coming from different servers, and we use two nanopublication monitors (in Zurich and Ottawa) to evaluate the responsiveness of the network.

In the second part of the evaluation we expose a server to heavy load from clients to test its retrieval capacity. For this we use a service called Load Impact (https://loadimpact.com) to let up to 100 clients access a nanopublication server in parallel. We test the server in Zurich over a time of five minutes under the load from a linearly increasing number of clients (from 0 to 100) located in Dublin. These clients are programmed to request a randomly chosen journal page, then to go though the entries of that page one by one, requesting the respective nanopublication with a probability of 10%, and starting over again with a different page. As a comparison, we run a second session, for which we load the same data into a Virtuoso SPARQL endpoint on the same server in Zurich (with 16 GB of memory given to Virtuoso and two 2.40 GHz Intel Xeon processors). Then, we perform exactly the same stress test on the SPARQL endpoint, requesting the nanopublications in the form of SPARQL queries instead of requests to the nanopublication server interface. This comparison is admittedly not a fair one, as SPARQL endpoints are much more powerful and are not tailor-made for the retrieval of nanopublications, but they provide nevertheless a valuable and well-established reference point to evaluate the performance of our system.

While the second part of the evaluation focuses on the server perspective, the third part considers the client side. In this last part, we want to test whether the retrieval of an entire dataset in a parallel fashion from the different servers of the network is indeed efficient and reliable. We decided to use a medium-sized dataset and chose LIDDI (*NP Index* RA7SuQ0e66, *2015*), which consists of around 100,000 triples. We tested the retrieval of this dataset from a computer connected to the internet via a basic plan from a regular internet service provider (i.e., not via a fast university network) with a command like the following:

```
$ np get -c -o nanopubs.trig RA7SuQ0e661LJdKpt5EOS2DKykf1ht9LFmNaZtFSDMrXg
```

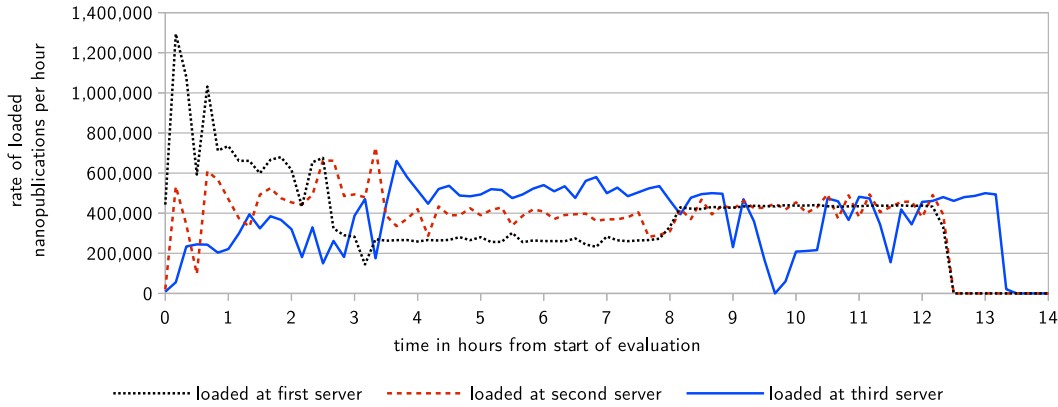

**Figure 6** **This diagram shows the rate at which nanopublications are loaded at their first, second, and third server, respectively, over the time of the evaluation.** At the first server, nanopublications are loaded from the local file system, whereas at the second and third server they are retrieved via the server network.

In addition, we wanted to test the retrieval in a situation where the internet connection and/or the nanopublication servers are highly unreliable. For that, we implemented a version of an input stream that introduces errors to simulate such unreliable connections or servers. With a given probability (set to 1% for this evaluation), each read attempt to the input stream (a single read attempt typically asking for about 8000 bytes) either leads to a randomly changed byte or to an exception being thrown after a delay of 5 s (both having an equal chance of occurring of 0.5%). This behavior can be achieved with the following command, which is obviously only useful for testing purposes:

```
$ np get -c -o nanopubs.trig --simulate-unreliable-connection \
    RA7SuQ0e661LJdKpt5EOS2DKykf1ht9LFmNaZtFSDMrXg
```

For the present study, we run each of these two commands 20 times. To evaluate the result, we can investigate whether the downloaded sets of nanopublications are equivalent, i.e., lead to identical files when normalized (such as transformed to a sorted N-Quads representation). Furthermore, we can look into the amount of time this retrieval operation takes, and the number of times the retrieval of a single nanopublication from a server fails and has to be repeated.

## Evaluation results

The first part of the evaluation lasted 13 h and 21 min, at which point all nanopublications were replicated on all three servers, and therefore the nanopublication traffic came to an end. Figure 6 shows the rate at which the nanopublications were loaded at their first, second, and third server, respectively. The network was able to handle an average of about 400,000 new nanopublications per hour, which corresponds to more than 100 new nanopublications per second. This includes the time needed for loading each nanopublication once from the local file system (at the first server), transferring it through the network two times (to the other two servers), and for verifying it three times (once when loaded and twice when received by the other two servers). Figure 7 shows the response times of the three servers as measured by the two nanopublication monitors in Zurich (top) and Ottawa (bottom)

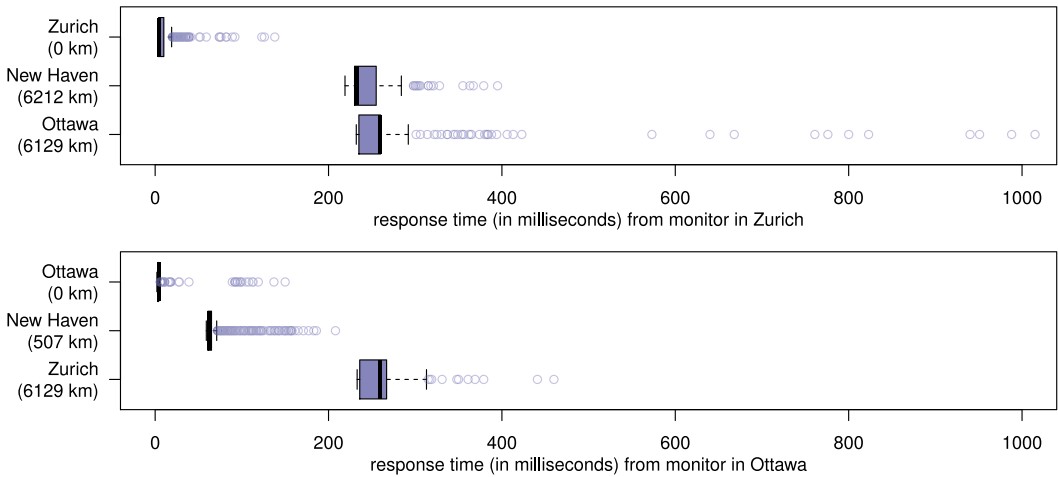

**Figure 7** Server response times under heavy load, recorded by the monitors during the first evaluation.

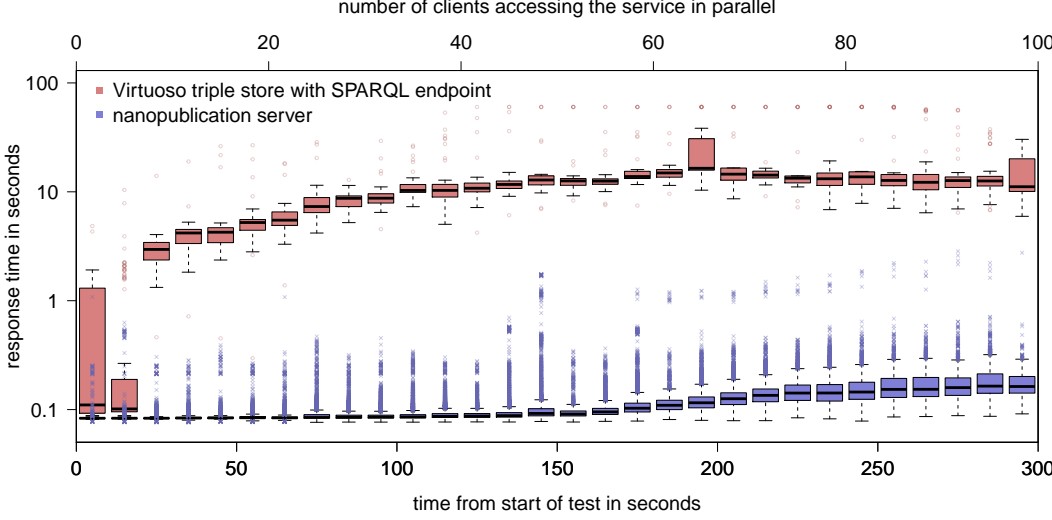

**Figure 8** Results of the evaluation of the retrieval capacity of a nanopublication server as compared to a general SPARQL endpoint (note the logarithmic *y*-axis).

during the time of the evaluation. We see that the observed latency is mostly due to the geographical distance between the servers and the monitors. The response time was always less than 0.21 s when the server was on the same continent as the measuring monitor. In 99.77% of all cases (including those across continents) the response time was below 0.5 s, and it was always below 1.1 s. Not a single one of the 4,802 individual HTTP requests timed out, led to an error, or received a nanopublication that could not be successfully verified.

Figure 8 shows the result of the second part of the evaluation. The nanopublication server was able to handle 113,178 requests in total (i.e., an average of 377 requests per second) with an average response time of 0.12 s. In contrast, the SPARQL endpoint answering the same kind of requests needed 100 times longer to process them (13 s on average),

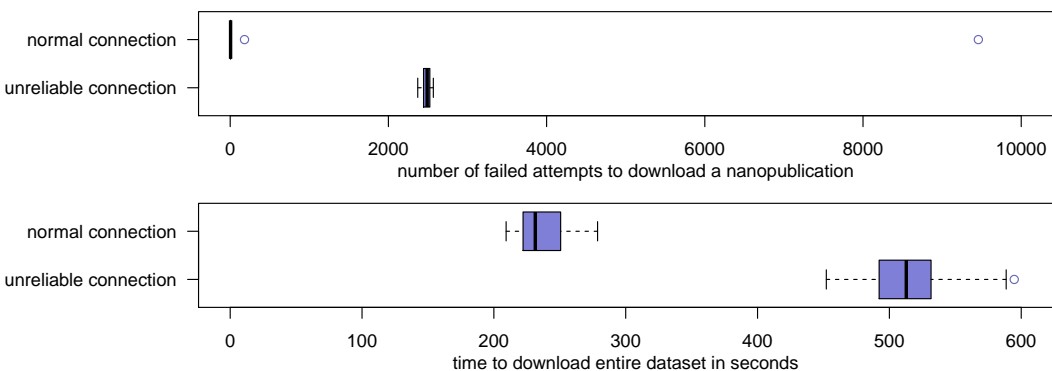

**Figure 9** The number of failures and required time when downloading the LIDDI dataset from the server network over a normal connection as well as a connection that has been artificially made unreliable.

consequently handled about 100 times fewer requests (1267), and started to hit the timeout of 60 s for some requests when more than 40 client accessed it in parallel. In the case of the nanopublication server, the majority of the requests were answered within less than 0.1 s for up to around 50 parallel clients, and this value remained below 0.17 s all the way up to 100 clients. As the round-trip network latency alone between Ireland and Zurich amounts to around 0.03 to 0.04 s, further improvements can be achieved for a denser network due to the reduced distance to the nearest server.

For the third part of the evaluation, all forty retrieval attempts succeeded. After normalization of the downloaded datasets, they were all identical, also the ones that were downloaded through an input stream that was artificially made highly unreliable. Figure 9 shows the number of retrieval failures and the amount of time that was required for the retrieval. With the normal connection, the downloading of nanopublications from the network almost always succeeded on the first try. Of the 98,184 nanopublications that had to be downloaded (98,085 content nanopublications plus 99 nanopublication indexes), fewer than 10 such download attempts failed in 18 of the 20 test runs. In the remaining two runs, the connection happened to be temporarily unreliable for "natural" reasons, and the number of download failures rose to 181 and 9,458, respectively. This, however, had no effect on the success of the download in a timely manner. On average over the 20 test runs, the entire dataset was successfully downloaded in 235 s, with a maximum of 279 s. Unsurprisingly, the unreliable connection leads a much larger average number of failures and retries, but these failures have no effect on the final downloaded dataset, as we have seen above. On average, 2,486 download attempts failed and had to be retried in the unreliable setting. In particular because half of these failures included a delay of 5 s, the download times are more than doubled, but still in a very reasonable range with an average of 517 s and a maximum below 10 min.

In summary, the first part of the evaluation shows that the overall replication capacity of the current server network is around 9.4 million new nanopublications per day or 3.4 billion per year. The results of the second part show that the load on a server when measured as response times is barely noticeable for up to 50 parallel clients, and therefore

the network can easily handle 50·$x$ parallel client connections or more, where $x$ is the number of independent physical servers in the network (currently $x = 10$). The second part thereby also shows that the restriction of avoiding parallel outgoing connections for the replication between servers is actually a very conservative measure that could be relaxed, if needed, to allow for a higher replication capacity. The third part of the evaluation shows that the client-side retrieval of entire datasets is indeed efficient and reliable, even if the used internet connection or some servers in the network are highly unreliable.

## DISCUSSION AND CONCLUSION

We have presented here a low-level infrastructure for data sharing, which is just one piece of a bigger ecosystem to be established. The implementation of components that rely on this low-level data sharing infrastructure is ongoing and future work. This includes the development of ''core services'' (see 'Architecture') on top of the server network to allow people to find nanopublications and ''advanced services'' to query and analyze the content of nanopublications. In addition, we need to establish standards and best practices of how to use existing ontologies (and to define new ones where necessary) to describe properties and relations of nanopublications, such as referring to earlier versions, marking nanopublications as retracted, and reviewing of nanopublications.

Apart from that, we also have to scale up the current small network. As our protocol only allows for simple key-based lookup, the time complexity for all types of requests is sublinear and therefore scales up well. The main limiting factor is disk space, which is relatively cheap and easy to add. Still, the servers will have to specialize even more, i.e., replicate only a part of all nanopublications, in order to handle really large amounts of data. In addition to the current surface feature definitions via URI and hash patterns, a number of additional ways of specializing are possible in the future: servers can restrict themselves to particular types of nanopublications, e.g., to specific topics or authors, and communicate this to the network in a similar way as they do it now with URI and hash patterns; inspired by the Bitcoin system, certain servers could only accept nanopublications whose hash starts with a given number of zero bits, which makes it costly to publish; and some servers could be specialized to new nanopublications, providing fast access but only for a restricted time, while others could take care of archiving old nanopublications, possibly on tape and with considerable delays between request and delivery. Lastly, there could also emerge interesting synergies with novel approaches to internet networking, such as Content-Centric Networking (*Jacobson et al., 2012*), with which—consistent with our proposal—requests are based on content rather than hosts.

We argue that data publishing and archiving can and should be done in a decentralized manner. We believe that the presented server network can serve as a solid basis for semantic publishing, and possibly also for the Semantic Web in general. It could contribute to improve the availability and reproducibility of scientific results and put a reliable and trustworthy layer underneath the Semantic Web.

### Funding

Ruben Verborgh is a postdoctoral fellow of the Research Foundation–Flanders (FWO). The other authors received no particular funding for this work. The funders had no role in study design, data collection and analysis, decision to publish, or preparation of the manuscript.

### Grant Disclosures

The following grant information was disclosed by the authors:
Research Foundation–Flanders (FWO).

### Competing Interests

Christine Chichester is an employee of Nestle Institute of Health Sciences, Lausanne, Switzerland. George Giannakopoulos is an employee of SciFY, a private not-profit company, Athens, Greece.

### Author Contributions

- Tobias Kuhn conceived and designed the experiments, performed the experiments, analyzed the data, wrote the paper, prepared figures and/or tables, performed the computation work, reviewed drafts of the paper.
- Christine Chichester conceived and designed the experiments, reviewed drafts of the paper.
- Michael Krauthammer, Núria Queralt-Rosinach, George Giannakopoulos and Axel-Cyrille Ngonga Ngomo performed the experiments, reviewed drafts of the paper.
- Ruben Verborgh and Raffaele Viglianti performed the experiments, wrote the paper, reviewed drafts of the paper.
- Michel Dumontier conceived and designed the experiments, performed the experiments, reviewed drafts of the paper.

### Data Availability

Source code:

- Java Library for Nanopublications: https://github.com/Nanopublication/nanopub-java
- Nanopublication server: https://github.com/tkuhn/nanopub-server
- Nanopublication monitor: https://github.com/tkuhn/nanopub-monitor

Content Data:

- All datasets: https://datahub.io/organization/nanopublications
- https://datahub.io/dataset/disgenet-v3-0-0-0-nanopubs
- https://datahub.io/dataset/nextprot-preliminary-nanopubs
- https://datahub.io/dataset/openbel-1-0-nanopubs
- https://datahub.io/dataset/openbel-20131211-nanopubs
- https://datahub.io/dataset/generif-aida-nanopubs
- https://datahub.io/dataset/linked-drug-drug-interactions-liddi

- https://datahub.io/dataset/disgenet-v2-1-0-0-nanopubs

Evaluation data:

- Repo with data and scripts of conference paper: https://bitbucket.org/tkuhn/trustypublishing-study/

- Repo with some additional content for this extended journal article: https://bitbucket.org/tkuhn/trustypublishingx-study/.

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
