# Peer review of "Decentralized provenance-aware publishing with nanopublications"

_PeerJ Computer Science, doi:10.7717/peerj-cs.78_

## Round 0.1 · original submission · Major Revisions

· Academic Editor

Major Revisions

The reviewers unanimously praise the submission for its clarity and interesting contribution. It is my view that, with suitable revisions, this article will be worthy of publication.

Two reviewers have flagged the originality of the submission with respect to the previous ISWC'15 publication. It is critical that the authors clarify, in their response and in the paper, the difference between this submission and their previous paper. There is an expectation that there must be a significant difference between the two papers.

I would expect that all comments raised by the reviewers are addressed and commented upon, one by one. I would however like to flag a few of them.

Some claims are made about nano-publications, such as they are never edited or they are of the same size. These claims need to be justified.

The authors claim some benefit of their approach, but these are not necessarily backed by empirical evaluation. In particular, issues around reliability and scalability have been flagged. I invite the authors to revisit their claim and/or evaluation section and ensure that claims are suitably evidenced by evaluation.

·

Basic reporting

In this paper the authors propose a low level infrastructure for distributing and decentralizing nanopublication datasets. As a part of the infrastructure, the authors also define tools for validating the nanopublications, publishing them and reviewing them. The approach is evaluated with 5 datasets spread around the globe.

The paper is well written and relevant for the Peerj journal, but I have several concerns with its originality and novelty. When I was reading the paper I had the feeling that I had already reviewed this work. By the time I reached section 2 I realized that indeed I had, because I was a reviewer of the ISWC paper from which this work is derived. I have thoroughly compared both versions, and my conclusion is that the contribution of this paper is exactly the same as the one described in the previous paper. More details are provided on this version (e.g., the part on how the servers propagate the nanopublication has been added), which is great (and I asked about this in my original review, so it's fantastic to see that the authors have acknowledged some of my comments), but I fail to see what does this work add to the previous original paper: the approach remains the same, the introduction and conclusions are almost a copy-paste of the original one and even the two evaluations are almost identical: table 1 has 2 more datasets but are not used in the evaluations, and figure 7 has been redrawn, but that's all. The web interface of the nanopublication validator is new, but that is not a significant contribution to become a new publication.

I already accepted the original paper because it was (and it is) a nice work. I am marking a "major review" decision because I want to give the authors and opportunity to explain to me why this paper is a significant contribution compared to the original one. As it stands now, I think that the paper should not be accepted in the journal.

Below I add additional comments from my original review which I think also apply to the current paper:

- The authors state that a nanopublication is never edited, just added. Then, what would happen if an author wanted to delete or retract a given published fact? How would the deletion affect the other copies among the server network?

- The data used for the evaluations i.e., for building figure 7 and 8 doesn't seem to be available. However the authors reference the datasets, so I don't think it is a major issue. Additionally, all the code is on Github, which is great.

Experimental design

All the comments regarding the originality of the research are included above. The methods and evaluations are clearly described in the paper.

Validity of the findings

The validity of findings is identical to the original (non extended) publication, because the evaluation is almost the same.

Reviewer 2 ·

Basic reporting

The authors adhere to the policies of the PeerJ Computer Science Journal.

Experimental design

The empirical evaluation conducted by the authors is clearly defined in my opinion. The setting used is clearly defined, and the results obtained are analysed in an objective manner.

Validity of the findings

The finding reported in the results are justified by the empirical evaluation. There is however the aspect related to reliability that has not been covered in the empirical validation (see comment 3 in the general comments to the authors).

Additional comments

The authors tackle in this paper the problem of publishing datasets in decentralized environments with distributed control over the publishing and the retrieval of datasets. The main motivation being the fact that there is currently no effective and reliable means for publishing datasets that are referenced from within scholarly publications.

The paper has the merit of reporting a practical solution. I enjoyed reading it as it is grounded with real examples all a long together with analyses that justify the choices made by the authors when elaborating their solution. That said, below are comments that I think are worthwhile addressing or at least discussing in the paper.

C1. The authors stated two main requirements: Having a reliable mechanism for hosting and referencing datasets, but also the ability to reference and retrieve datasets at different granularity levels. With respect to the second requirements, the authors did not discuss the process by which a datasets is transformed into a set of nanopublications. The model proposed by the authors support nano-publication and nano-publication indexes, which references other nano-publication. The question the reader may ask is how nan-publication and their container nano-publications indexes are obtained given a dataset.

C2. A second aspect that is related to the first comment is the heterogeneity of datasets. Scientific datasets are usually heterogeneous, and the most part of it is not stored in the form of RDF, but instead in CSV, relational, etc. Can the solution proposed by the authors cater for this kind of datasets?

C3. Regarding reliability, it is not clear on how the dataset is replicated among the server. In other words, what is the replication scheme used to ensure the availability of the datasets.

C4. I also note that in the evaluation section, the problem of reliability has not been examined.

C5. The publication of datasets should in general be accompanied with the publication of metadata within catalogues that inform prospective users with their availability. I think that this aspect is worth discussing in the paper.

Minor comments.

C6. At the end of the introduction, the authors state that the article is an extended version of a previous article. The authors need to state clearly what is new in the paper submitted to the PeerJ.

C7. The URIs used by the authors contain the artifact code which obtained by analyzing the data content. This raises the question as to the cost that this operation may incur when building the URIs. This aspect was touched on in the paper, but briefly, and need further discussion.

C8 In Figure 2 and the text that explain the Figure, the author use the expression “propagation”. Later this term is explained, but until then it is not clear for the reader what is meant by propagation.

C9. In page 5, 1st paragraph towards the end: reminder -> remainder

Reviewer 3 ·

Basic reporting

The submission is very clearly written and well presented. It is mostly self-contained, although some aspects could benefit from some additional background information.

1. Although past literature regarding trusty URIs was cited, I did not understand how they fit into the nanopublication server network (section 3.2) until I re-read the paper from ESWC. So the idea is that the proposed approach prerequisites the ability of having trusty URIs for all the nanopublications exchanged within the network. It is kind of what figure 2 shows and what section 3.2 says. But could have been made clearer. Also, does this mean that all the nanopublications exchange in this network must be published using the nanopublication java library, or something similar? Then I would say the schematic architecture is a bit too high level for readers to understand these key points. A technical architecture diagram, at least for representing an instance implementation of the proposal, could be helpful.

2. On line 257 the authors claim that there is an assumption that all nanopublications should be all similar in size, which makes it one of the advantages of the proposed approach. As someone who do know about nanopublications, I am not sure this is a universal true. Have the authors got some citations to support this statement, or is this a kind of best practices expected from nanopublication users?

Experimental design

The hypothesis of the submission is: Can we create a decentralized,  reliable, trustworthy, and scalable system for publishing, retrieving, and archiving datasets in the form of  sets of nanopublications based on existing Web standards and infrastructure? Therefore, we expect the experiment to evaluate the following aspects of the proposed system: its reliability, trustworthiness, and scalability. However, in my opinion the experiments are not completely satisfactory.

In terms of evaluation1, it does not seem ideal to take more than 13 hours to load and populate the test datasets. Obviously it will probably take marginal time to load and populate a singular nanopublication. But as the authors say, it’s often that people would publish a collection of nanopublications as a dataset. What would be the typical size then? Would make any sense to evaluate the load time for different sized nanopublication datasets? I would say 13 hours is a very long time even for 200 million triples.

In terms of the second evaluation, what is the SPARQL query used for testing the Virtuoso server? The experiment could use more explanation. Currently, the procedure seems to be very much tailored to the described system, for example, going through the internal journal page etc. Is this a realistic nanopublication retrieval scenario? Also, this did not demonstrate retrieval in a decentralized setting, but simply a stress testing of a single server.

The evaluation section seems a bit incomplete and weak. How about scalability? What would be the scalability bottleneck of the proposed design, the indexes or the propagation procedure? Could the system handle communication of 100 or several hundred nodes?

Validity of the findings

The data could be better explained so that users can understand why something worked well (such as the reliability line 603-617), and how they support the claimed hypothesis. If evaluations are incomplete in terms of supporting the whole hypothesis (such as trustworthiness), then the authors should explain whether the evaluation may be incomplete.

---

## Round 0.2 · Minor Revisions

· Academic Editor

Minor Revisions

Please address the remaining comments raised by reviewers 1 and 2.

·

Basic reporting

The authors seem to have addressed all my comments successfully. They have shown how the paper includes new material from the former ISWC publication, and I think that the response for being able to edit or fix nanopublications makes a lot of sense. I would have referred to that more as "archival" rather than publishing, but I don't think it's a big issue.

What I would like to recommend the authors is to include the link to the repository with the data from the evaluations (https://bitbucket.org/tkuhn/trustypublishing-study/src/). That way anyone interested in the data would be able to access it.

Experimental design

The authors have expanded on the evaluation and discussed some limitations of their approach

Validity of the findings

No comments

Additional comments

No comments

Reviewer 2 ·

Basic reporting

The article is well written and make a pragmatic proposal that I think is useful and interesting for the linked data community.

Experimental design

The authors addressed my main concern about the empirical evaluation by running a new experiment to assess the reliability aspect.

Validity of the findings

The ideas presented in the paper are backed by empirical evaluation, and as such I think that the findings are solid.

Additional comments

I would like to thank the authors for attempting to address my comments. They have managed to address most of them, and ran a new experiment, albeit of a small size, to address the point of reliabaility that I raised. There are 2 comments from the previous round that still need to be addressed by the authors.

C1. The authors did not address my first comment in the previous round. “The authors stated two main requirements: Having a reliable mechanism for hosting and referencing datasets, but also the ability to reference and
retrieve datasets at different granularity levels. With respect to the second
requirements, the authors did not discuss the process by which a datasets is
transformed into a set of nanopublications. The model proposed by the authors
support nano-publication and nano-publication indexes, which references other
nano-publication. The question the reader may ask is how nan-publication and
their container nano-publications indexes are obtained given a dataset”.

C2. In their response to C2 in the previous round, the authors suggest that heterogeneity of the data model used in datasets (e.g., CSV and relational) can be resolved by using existing state of the art technique to translate data in those models into RDF. This approach may be expensive, I was wondering if lightweight approaches which do not attempt to translate the original data, but instead create metadata that describe them using nanopublication would be more realistic and cost effective as a solution. Of course the granularity of retrieval in this case would be a whole dataset, but there are scenarios where this solution would be acceptable. I think that a discussion in these lines that clariy the options to the reader would be helpful, specially that linked data form only a small proportion of available scientific data.

Reviewer 3 ·

Basic reporting

No comments

Experimental design

A new evaluation is added to strengthen evaluation.

Validity of the findings

Hypothesis has been adjusted according to the findings.

Additional comments

I am happy with the responses from the authors and would therefore recommend acceptance of its publication.

---

## Round 0.3 · accepted · Accept

· Academic Editor

Accept

It is my pleasure to accept your submission. You have satisfactorily addressed the remaining comments. Congratulations!

---

## Author Rebuttal · Round 0.3

Below are our responses to the specific points raised by the editor and the reviewers.

> Editor's comments

> Please address the remaining comments raised by reviewers 1 and 2

Done. See below.

> Reviewer 1 (Daniel Garijo)

> What I would like to recommend the authors is to include the link to the
> repository with the data from the evaluations
> (https://bitbucket.org/tkuhn/trustypublishing-study/src/). That way anyone
> interested in the data would be able to access it.

That is a very good idea. The links to the two repositories are now included in
the beginning of the evaluation section.

> Reviewer 2 (Anonymous)

> C1. The authors did not address my first comment in the previous round. "The
> authors stated two main requirements: Having a reliable mechanism for hosting
> and referencing datasets, but also the ability to reference and retrieve
> datasets at different granularity levels. With respect to the second
> requirements, the authors did not discuss the process by which a datasets is
> transformed into a set of nanopublications. The model proposed by the authors
> support nano-publication and nano-publication indexes, which references other
> nano-publication. The question the reader may ask is how nan-publication and
> their container nano-publications indexes are obtained given a dataset".

We now understand your point about data being represented in RDF but not in
nanopublication format. We previously misunderstood it as being covered by our
response to C2. To address it, we added a few sentences to the first paragraph
of the approach section, sketching the problem and a starting point for its
solution:

  "We furthermore exploit the fact that datasets in RDF can be split into small
  pieces without any effects on their semantics. After Skolemization of blank
  nodes, RDF triples are independent and can be separated and joined without
  restrictions. Best practices of how to define meaningful small groups of such
  triples still have to emerge, but an obvious and simple starting point is
  grouping them by the resource in subject position. We focus here on the
  technical questions and leave these practical issues for future research."

With respect to the generation of nanopublication indexes, we now explain this
explicitly in the end of the section Nanopublication Indexes, and refer to the
subsequent section, which explains how the nanopublication library can be used
for this.

> C2. In their response to C2 in the previous round, the authors suggest that
> heterogeneity of the data model used in datasets (e.g., CSV and relational)
> can be resolved by using existing state of the art technique to translate data
> in those models into RDF. This approach may be expensive, I was wondering if

> lightweight approaches which do not attempt to translate the original data,
> but instead create metadata that describe them using nanopublication would be
> more realistic and cost effective as a solution. Of course the granularity of
> retrieval in this case would be a whole dataset, but there are scenarios where
> this solution would be acceptable. I think that a discussion in these lines
> that clariy the options to the reader would be helpful, specially that linked
> data form only a small proportion of available scientific data.

This is a very good point. Thank you for raising it. We have in fact been
thinking about such light-weight dataset announcements, in particular by using
the HCLS Community Profile for dataset descriptions. We now include the
following paragraph at the end of the section on nanopublication indexes:

  "As a side note, dataset metadata can be captured and announced as
  nanopublications even for datasets that are not (yet) themselves available in
  the nanopublication format. The HCLS Community Profile of dataset descriptions
  (Gray et al., 2015) provides a good guideline of which of the existing RDF
  vocabularies to use for such metadata descriptions.